# The role of renal and liver function in clinical ctDNA testing

Jens Bo Koudahl Conrad[1,2], Tenna Vesterman Henriksen[1,2], Jesper Berg Nors[1,2,3], Mads Heilskov Rasmussen[1,2], Mai-Britt Worm Ørntoft[1,2,4], Nis Hallundbæk Schlesinger[5], Per Vadgaard Andersen[6], Kåre Andersson Gotschalck[2,7], Claus Lindbjerg Andersen[1,2] *

1 Department of Molecular Medicine, Aarhus University Hospital, Aarhus, Denmark, 2 Department of Clinical Medicine, Aarhus University, Aarhus, Denmark, 3 Department of Surgery, Randers Regional Hospital, Randers, Denmark, 4 Department of Surgery, Gødstrup Regional Hospital, Gødstrup, Denmark, 5 Department of Surgery, Bispebjerg Hospital, Copenhagen, Denmark, 6 Department of Surgery, Odense University Hospital, Odense, Denmark, 7 Department of Surgery, Horsens Regional Hospital, Horsens, Denmark

* cla@clin.au.dk

## Abstract

Circulating tumor DNA (ctDNA) has high clinical potential in early cancer detection. The renal system and the liver are involved in clearing circulating cell free DNA (cfDNA) from the blood. Recent studies on mice show that inhibiting the liver's ability to clear cfDNA results in elevated ctDNA levels in blood samples. Emphasizing the need for studies in humans exploring if markers of renal and liver function are associated with cfDNA and ctDNA levels in the blood. The present study investigates if cfDNA level, ctDNA level and ctDNA detection is affected in colorectal cancer (CRC) patients with clinical biomarkers indicative of low renal and liver function. We requisitioned standard laboratory tests of renal and liver function, measured within thirty days of curative intended surgery from 846 stage I-III CRC patients. For each patient, matching preoperative cfDNA and ctDNA data was available. We investigated the correlation between impaired renal and liver function and cfDNA level, ctDNA level, and ctDNA detection. The findings revealed that variation in renal and liver function in stage I-III CRC patients did not affect cfDNA level, ctDNA level, or ctDNA detection and that ctDNA test results remain stable over a wide range of renal and liver biomarker results.

## 1. Introduction

Measurements of circulating tumor DNA (ctDNA) is an emerging tool for early cancer detection, allowing for in-depth monitoring of treatment effect and disease progression [1–5]. Despite ctDNA detection technologies becoming increasingly sensitive, ctDNA is not detectable in all cancer patients. The ability to detect ctDNA is dependent on the level of ctDNA in the blood, which again is determined by the release and clearance-rate of tumor DNA [6]. The supply of ctDNA to the blood is driven by cancer cell turnover [7], which is closely linked to disease stage and location [8]. In healthy individuals, the half-life of ctDNA in the circulation is estimated to ~ 2 hours [8]. Circulating cell-free DNA (cfDNA) in blood has been reported to be cleared via the liver and kidneys [9]. In support of this, increased cfDNA levels have

**Data availability statement:** Because the data contains sensitive personal patient information, it cannot be made publicly available. Access to data requires that the data requestor (legal entity) enter into Collaboration and Data Processing Agreements, with the Central Denmark Region (the legal entity controlling and responsible for the data). Inquiries for access can be addressed to the Data Access Committee at Department of Molecular Medicine, Aarhus University Hospital (contact via moma@rm.dk).

**Funding:** This study was funded by a scholar-stipend from the Danish Independent Research Fund (Conrad) and support from the Novo Nordisk Foundation [grant number NNF22OC0074415 (Andersen)], Innovation Fund Denmark [grant number 9068-00046B (Andersen)] and the Danish Cancer Society [grant numbers R231-A13845, R257-A14700, R352-A20664 (Andersen)]. The funding organizations and entities listed above had no influence on the study's design, data collection, analysis, interpretation of the data, manuscript preparation and review, or decision to submit for publication.

**Competing interests:** The authors have declared that no competing interests exist.

been reported in patients with poor renal [10–12] and liver function [13,14]. Furthermore, it was recently reported that transiently blocking the liver cfDNA clearance function in mice increased the cfDNA level in circulation by up to 10-fold, and that this in turn increased the sensitivity for detecting ctDNA in mice with small tumors [15].

Patients afflicted with cancer disproportionately suffer from liver and renal diseases when compared to the background population. In patients with non-alcoholic fatty liver disease the risk of colorectal cancer (CRC) is elevated 3-fold [16] and in patients with CRC abnormally low renal function has an estimated prevalence of over 70% [17]. Furthermore, the presence of a cancer in the body may in and of itself degrade the condition and function of the liver and kidneys [18–20]. Hence, it may be hypothesized that CRC patients affected by poor liver and renal function may have attenuated ctDNA clearance, and therefore augmented levels of ctDNA. Potentially, ctDNA testing will be more sensitive in these patients than in patients without liver and kidney comorbidities.

CRC patients in Denmark undergo a range of laboratory tests prior to curatively intended surgery. These include creatinine, glomerular filtration rate (eGFR), sodium, and potassium. Collectively, these parameters estimate the fluid status and kidney function of a patient; Indeed, estimates of eGFR form the foundation for diagnosing patients with chronic kidney disease. Liver function is estimated via blood tests of bilirubin, alkaline phosphatase, and alanine transaminase among others. The laboratory test data is archived and readily available, providing a detailed view into the renal and liver function of CRC patients in the Danish healthcare system. To investigate the impact of renal and liver function on cfDNA level, ctDNA level, and ctDNA detection probability, we analyzed a cohort of CRC patients for which preoperative liver and renal function laboratory tests as well as information of cfDNA and ctDNA level was available.

## 2. Methods

### 2.1. The patient cohort

The cohort for this study comprised consecutively recruited patients planned to receive a curative intended surgical resection of stage I-III CRC. All patients were diagnosed and treated in accordance with Danish guidelines (DCCG.dk). The participants were recruited from 2012 to 2022 at nine hospitals in the Capital, Central and Southern Denmark regions. As part of the study, patients had blood collected prior to and after surgery with the aim to retrospectively measure cfDNA and ctDNA. All patients gave written informed consent in accordance with the World Medical Association Declaration of Helsinki. The project was approved by The Central Denmark Committees on Health Research Ethics (case j. no. 1-10-72-3-18 and j. no. 1-10-72-223-14).

### 2.2. Extracting data on laboratory test results

Laboratory test results of renal and liver function were requisitioned from the local "Clinical Laboratory Information System Research Databases" (LABKA) in each of the involved Danish regions. To gauge liver function, alkaline phosphatase, bilirubin and alanine transaminase results were requisitioned. To gauge renal function, creatinine, eGFR, sodium, and potassium results were chosen. The LABKA databases do not report precise eGFR results above $90\,mL/min/1.73m^2$. Therefore, any eGFR test result of $>90\,mL/min/1.73m^2$ was registered as $90\,mL/min/1.73m^2$ in this study.

### 2.3. Classification of laboratory test results

The laboratory results were categorized according to the reference range for each sample. In Denmark, the hospital system is organized in five independent regions, each of which

define the normal reference range for their laboratories. The reference range for a specific patient's test result was defined using the reference range specific to the Region the patient was recruited from, accounting for age and sex where applicable [21–23] (S1 Table in S1 Text). Based on the reference ranges the results for each patient were categorized as "Below reference", "Within reference", or "Above reference".

## 2.4. Tumor and cfDNA

The ctDNA quantification data included in this paper have been previously published in their own right [24]. Detailed methodology is described there and in the associated methodology papers (24-28). In brief, whole-exome sequencing was conducted on paired tumor and buffy coat DNA samples from each patient. This data was evaluated for clonal mutations, selected based on variant allele frequencies and estimates of cancer cell purity, tumor ploidy, and allele-specific copy numbers [25]. Based on the available clonal mutations, ctDNA analysis was conducted using either droplet digital PCR targeting a single, clonal, small-nucleotide variant [26,27] or deep targeted panel sequencing of 12 genes frequently mutated in CRC [28].

## 2.5. Pairing biomarkers of renal and liver function to ctDNA test results

Laboratory test results were matched with the preoperative and postoperative cfDNA samples. For patients with repeated laboratory measurements, the measurement closest in time to the cfDNA blood collection was chosen. Only test results from within 30 days before the operation were considered when matching results to preoperative ctDNA and cfDNA measurements.

## 2.6. Defining low renal function

In this study, we define a 30-day preoperative eGFR as the mean of all eGFR measurements conducted in the month leading up to the operation. The formulas used by the regional hospitals to calculate eGFR is available in S1 File. The median number of eGFR measurements per patient in the month up to operation was 2 (IQR 1-3). The 30-day preoperative eGFR was used to define whether a patient had low renal function or normal renal function. In accordance with Kidney Disease: Improving Global Outcomes (KDGIO) guidelines (https://kdigo.org) patients were categorized as having low renal function if their average eGFR was below 60 mL/min/1.73m2

## 2.7. Statistical analysis

The ctDNA status (detected or undetected), ctDNA level, and cfDNA level were the dependent variables whereas the laboratory test results were the primary explanatory variables. To investigate the relationship between the liver function and the risk of a positive ctDNA call, binomial logistic regression analysis was applied. The association between renal function and ctDNA test results was similarly analyzed with a binomial logistic regression. A log-log regression was used to analyze the relationship between the liver/renal laboratory test level and the ctDNA or cfDNA level. Only ctDNA positive patients were included in the log-log regression analysis. All regression models were adjusted for patient age at date of operation and the pT and pN categories of the tumor. In all analyses, the false discovery rate was controlled using the Benjamini-Hochberg method. A significant result was defined as $p < 0.05$ after Benjamini-Hochberg adjustment. All statistical analysis was conducted using the programming language R, version 4.3.1.

## 3. Results

In total, we included 846 stage I-III CRC patients. Patient characteristics are presented in Table 1. We measured cfDNA and ctDNA preoperatively on all patients. Clinical routine

biomarker levels indicative of renal function and liver function were available on different subsets of patients (S2 Table in S1 Text).

### 3.1. Association between renal function and cfDNA and ctDNA

To explore the impact of renal function on ctDNA and cfDNA test results, we investigated whether the sodium, potassium, creatinine, and eGFR levels were correlated to cfDNA level, ctDNA level, and ctDNA detection status. None of the renal biomarkers were significantly associated with cfDNA level, ctDNA level, or ctDNA detection status (Fig 1). Full regression results are available in S3–S6 Tables in S1 Text.

We further investigated whether patients with normal and low renal function differed in their cfDNA and ctDNA levels (Fig 2). A correlation between increased cfDNA level and low renal function (OR = 2.415 p = 0.007) was observed. However, after adjusting for patient age and disease stage the association disappeared (OR = 1.380 p = 0.369) (Fig 2A). No association was observed between ctDNA level and renal function neither with or without adjustments for age and low renal function were observed (Fig 2B). Full regression results are available in S7 Table in S1 Text.

### 3.2. Association between liver function and cfDNA and ctDNA

To explore the possible link between liver function and ctDNA and cfDNA levels, we analyzed whether the bilirubin, alanine transaminase, and alkaline phosphatase were correlated to cfDNA level, ctDNA level, and ctDNA detection rate (Fig 3). We found that elevated bilirubin levels were correlated with increased cfDNA levels (R = 0.148, p = 0.022). However, there were no statistically significant correlations to ctDNA level or ctDNA detection status. Neither alkaline phosphatase nor alanine transaminase were significantly associated with cfDNA and ctDNA levels nor with ctDNA detection status. Full regression results are available in S3–S6 Tables in S1 Text.

**Table 1. Clinical characteristics of the study cohort.**

| Characteristic | N = 846[1] |
|---|---|
| Age | 71 (64, 77) |
| Sex | |
| Female | 383 (45%) |
| Male | 463 (55%) |
| T stage | |
| pT1 | 35 (4.1%) |
| pT2 | 90 (11%) |
| pT3 | 635 (75%) |
| pT4 | 86 (10%) |
| N stage | |
| pN0 | 569 (67%) |
| pN1 | 180 (21%) |
| pN1c | 6 (0.7%) |
| pN2 | 91 (11%) |
| Location | |
| Left side | 420 (50%) |
| Right side | 387 (46%) |
| Rectum | 39 (4.6%) |
| [1]Median (IQR); n (%) | |

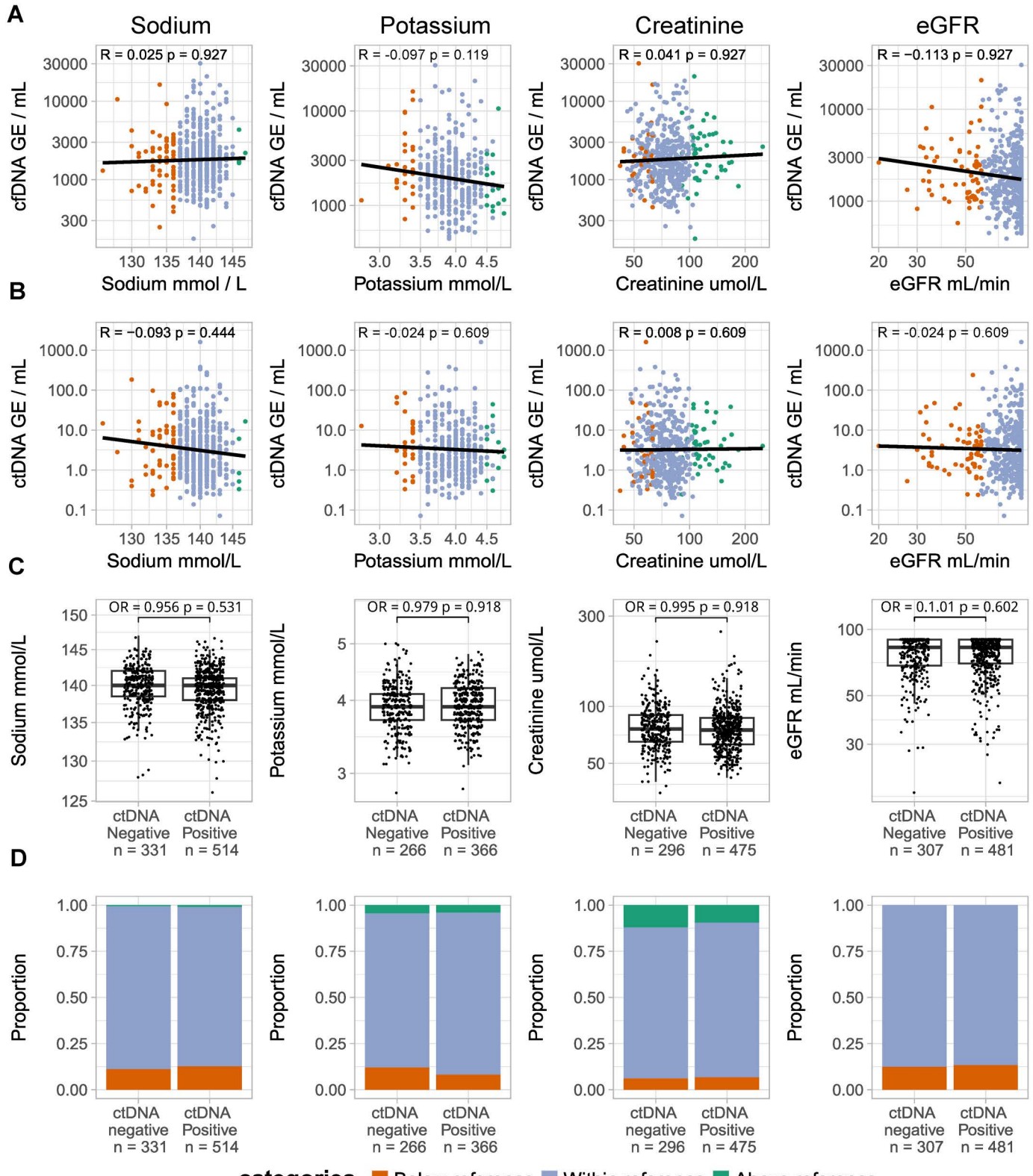

**Fig 1. Association between liver function tests and the cfDNA level, ctDNA level, and ctDNA detection.** (A) Scatterplots comparing cfDNA level (top) and ctDNA level (bottom) to renal function test results. (B) Scatterplots comparing ctDNA level to renal function test results. In A and B dots are colored for the reference range categories. A log-log regression was used to analyze the relationship between the laboratory test results and the ctDNA or cfDNA level. (C) Box

plots depicting test results in ctDNA negative and ctDNA positive patients. The relationship between the laboratory test result and the risk of a positive ctDNA call was analyzed with a binomial logistic regression. (D) Stacked bar chart showing the proportion of patients in each reference range category in ctDNA negative and ctDNA positive patients. For eGFR, a result above 90 mL/min/1.73m2 was treated as 90 mL/min/1.73m2. All regression models were adjusted for patient age at date of ctDNA sampling, and the pT and pN categories of the tumor. In all analyses, the false discovery rate was controlled using the Benjamini-Hochberg method. A significant result was defined as **p** < 0.05 after Benjamini-Hochberg adjustment. Note that different patients can have different reference ranges depending on age, sex, and region of origin. Therefore, the same laboratory result can be categorized in different reference range categories.

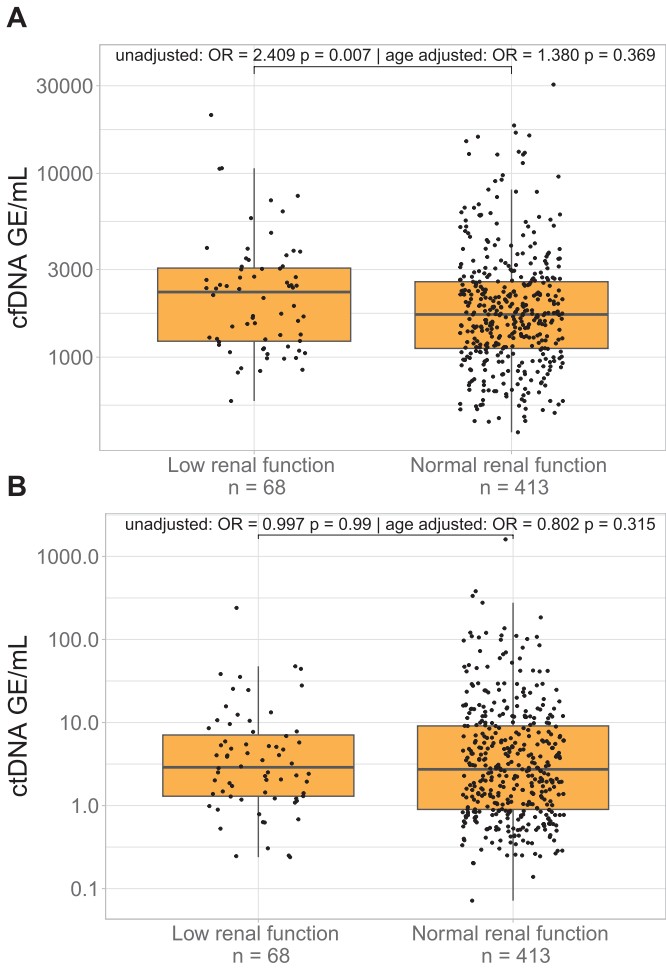

**Fig 2. Box plot showing the cfDNA level (A) and ctDNA level (B) in patients with low renal function and patients with normal renal function.** The association was assessed using binomial logistic regression. The statistical analysis was adjusted for pT stage and pN stage as well as age at sample date.

## 4. Discussion

In this study, we investigated whether preoperative laboratory biomarkers of renal and liver function were correlated to preoperative cfDNA level, ctDNA level, and ctDNA detection in CRC patients. We paired cfDNA test results to data from the Danish healthcare system's LABKA databases.

We observed that patients with higher bilirubin levels tended to have higher levels of cfDNA, though the association was minor. However, the increased cfDNA levels did not

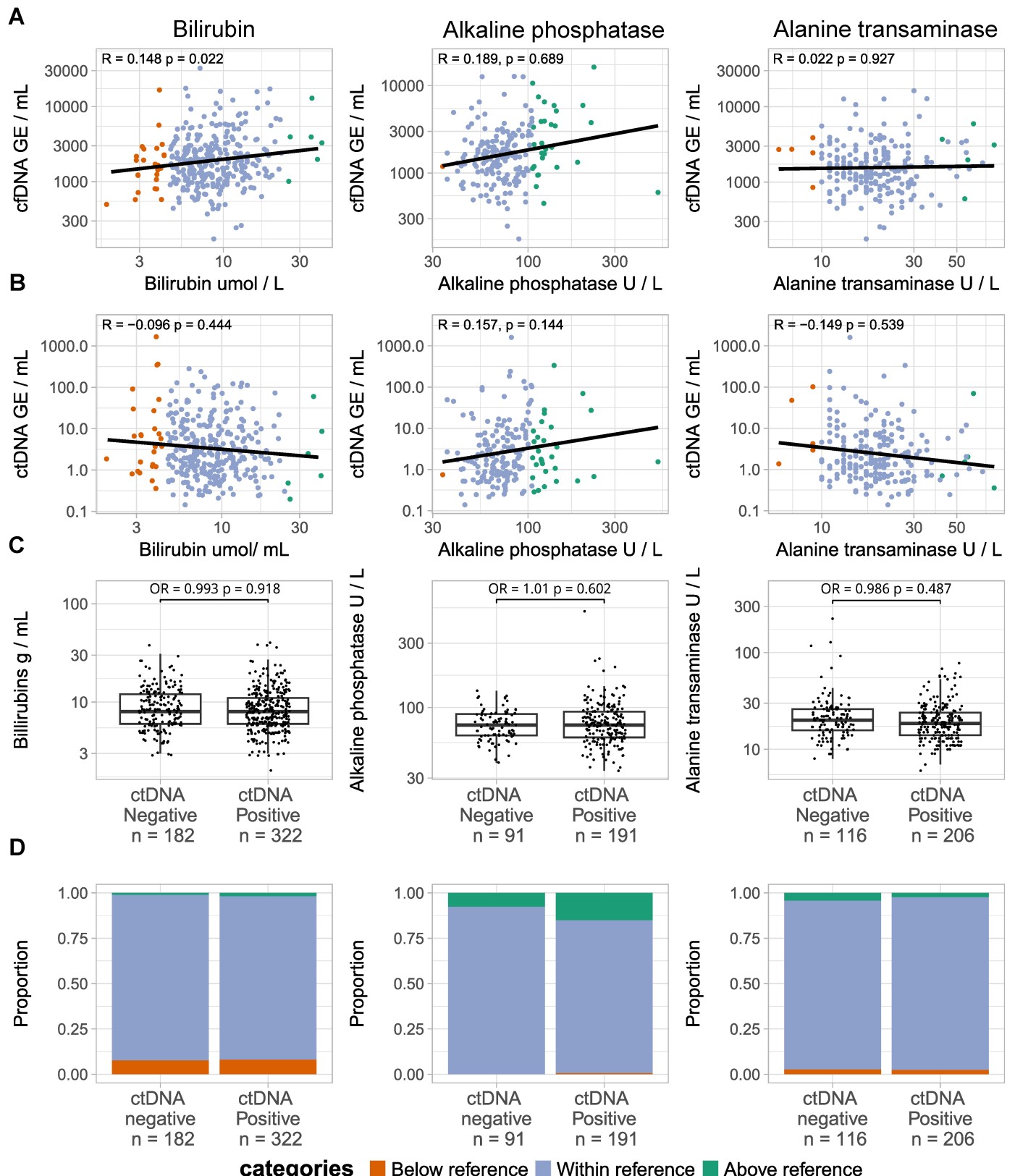

**Fig 3. Association between liver function tests and the cfDNA level, ctDNA level, and ctDNA detection.** (A) Scatterplots comparing cfDNA level (top) and ctDNA level (bottom) to liver function test results. (B) Scatterplots comparing ctDNA level to liver function test results. In A and B dots are colored for the reference

range categories. A log-log regression was used to analyze the relationship between the laboratory test results and the ctDNA or cfDNA level. (C) Box plots depicting test results in ctDNA negative and ctDNA positive patients. The relationship between the laboratory test result and the risk of a positive ctDNA call was analyzed with a binomial logistic regression. (D) Stacked bar chart showing the proportion of patients in each reference range category in ctDNA negative and ctDNA positive patients. All regression models were adjusted for patient age at date of ctDNA sampling, and the pT and pN categories of the tumor. In all analyses, the false discovery rate was controlled using the Benjamini-Hochberg method. A significant result was defined as **p** < 0.05 after Benjamini-Hochberg adjustment. Note that different patients can have different reference ranges depending on age, sex, and region of origin. Therefore, the same laboratory result can be categorized in different reference range categories.

translate to increased ctDNA levels or ctDNA detection rates; if increased bilirubin levels were indicative of decreased cfDNA clearance, we would expect the ctDNA level to increase as well. None of the other investigated markers showed statistically significant associations with cfDNA or ctDNA test results. We did find that patients with low renal function had increased levels of cfDNA, but this association lost statistical significance when the analysis was adjusted for patient age. This likely reflects that advanced age both associated with increased cfDNA levels and lower renal function [29,30]. In addition we conducted a supplementary investigation of cfDNA levels in the postoperative setting (S2 File) and found no significant association between postoperative cfDNA level and biomarkers of renal and liver function.

Our results suggest that CRC patients eligible for curative intended surgery do not experience abnormal ctDNA and cfDNA clearance based on their estimated renal and liver function. The vast majority of the standard renal and liver marker laboratory test results were within the normal reference ranges. One can expect that an effect on cfDNA clearance would be more apparent in patients with severe liver disease and end-stage renal disease. Indeed, in patients with nonalcoholic fatty liver disease, Karlas et al. found that disease severity correlated with higher cfDNA levels [14]. Additionally, Coimbra et al. found that cfDNA was elevated in patients who suffered from end-stage renal disease [11]. It may therefore be that the liver and renal function of the included CRC patients is not low enough to affect ctDNA clearance. Patients with much lower renal and liver function, than those included in this study, can be found in Denmark, but these patients would very rarely be candidates for curative intended surgery in the Danish healthcare system.

Although our results indicate that ctDNA clearance inhibition does not occur naturally in stage I-III CRC patients, Martin-Alonso et al. [15] found that blockage of liver-mediated cfDNA clearance in mice resulted in increased ctDNA concentrations. If also effective in humans, this could have great implications for future ctDNA testing, as low ctDNA concentrations is the most challenging aspect of ctDNA detection. However, if liver-based ctDNA clearance was already inhibited in a subset of CRC patients, the ctDNA priming strategy based on inhibiting liver clearance might have limited effect. Our observation that ctDNA levels are stable across the range of liver biomarkers indicate that this should not be a concern.

While ctDNA detection is primarily utilized as a strong predictor of cancer relapse, ctDNA level in the blood and its growth rate also has utility as a prognostic marker [27]. The variable ctDNA levels detected in blood samples remains a challenge for the field though [8]. Improving our understanding of what factors determine ctDNA level in patients will allow researchers to better account for biological variation when they use ctDNA level as a prognostic indicator. Indeed, much work is being done identifying which patient parameters influence ctDNA availability [31–33].

While using hospital databases to gather data on blood-based clinical biomarkers allowed for a highly representative dataset on a large cohort of CRC patients, it comes with some limitations. Most of the clinical biomarkers were collected on the discretion of attending clinicians. The availability of some biomarkers will therefore be subject to confounding by indication.

Additionally, when using the laboratory databases, analyses of continuous measures were limited for markers with values below or above the quantifiable range. Our study focused on laboratory test results in the short 30-day period before curative intended surgery as this period is the standard length of the Danish preoperative CRC cancer work-up program. However, this has the limitation of not meeting KDGIOs recommended 3-month sampling period for assessing long-term renal function. While most of our patients had multiple eGFR measurements in the 30-day period, a minor fraction had just a single eGFR measurement (n = 181 37.6%). These singular measurements provide important information, yet will not account for the patient's biological variance in eGFR. Finally, ctDNA samples and clinical biomarker samples were not always collected on the same day, which may introduce some variance.

In summary, this study indicates that variation in liver and renal function among patients with localized CRC does not influence ctDNA level or ctDNA detection. While more research is required to elucidate factors governing ctDNA release and clearance, implementation of ctDNA priming strategies in a human setting could potentially increase ctDNA levels, thus radically increasing ctDNA test sensitivity. For now, it is encouraging that abnormal liver and renal function does not need to be considered when conducting ctDNA testing, as the ctDNA clearance can be considered robust across a wide range of test results in CRC patients undergoing curative intended treatment.

## Supporting Information

**S1 Fig. Association of eGFR with cfDNA level stratified by age.** Figures showing grouped eGFR levels and continuous cfDNA levels stratified for patient age. Associations were calculated with a paired Wilcoxon signed rank test.
(EPS)

**S2 Fig. Association of eGFR with ctDNA level stratified by age.** Figures showing grouped eGFR levels and continuous ctDNA levels stratified for patient age. Associations were calculated with a paired Wilcoxon signed rank test.
(EPS)

**S1 Text. S1 Table.** Reference ranges for the laboratory tests. **S2 Table.** Number of patients for whom a specific laboratory test was available. **S3 Table.** Full log-log linear regression results for cfDNA. **S4 Table.** Full log-log linear regression results for ctDNA. **S5 Table.** Full binomial logistic regression results for ctDNA detection compared to laboratory test result. **S6 Table.** Full binomial logistic regression results for ctDNA detection compared to reference range category of laboratory test. **S7 Table.** Full binomial logistic regression results for ctDNA and cfDNA compared to low and high renal function.
(XLSX)

**S1 File. Description of eGFR calculation formulas.**
(DOCX)

**S2 File. Postoperative analysis of cfDNA and laboratory biomarkers of renal and liver function.**
(DOCX)

**S3 Fig. Potassium_chart.**
(EPS)

**S4 Fig. Postop_cohort_funnel.**
(EPS)

**S5 Fig. Postop_cfDNA_linear_regression.**
(EPS)

**S6 Fig. Postop_renal_function.**
(EPS)

## Acknowledgements

We thank the blood donors, CRC patients, and The Danish Cancer Biobank for contributing clinical material. The study was conducted as part of the Danish National Center for Circulating Tumor DNA Guided Cancer Treatment.

## Author contributions

**Conceptualization:** Jens Bo Koudahl Conrad, Tenna Vesterman Henriksen, Kåre Andersson Gotschalck, Claus Lindbjerg Andersen.

**Data curation:** Jens Bo Koudahl Conrad, Tenna Vesterman Henriksen, Nis Hallundbæk Schlesinger, Per Vadgaard Andersen, Kåre Andersson Gotschalck, Claus Lindbjerg Andersen.

**Formal analysis:** Jens Bo Koudahl Conrad.

**Funding acquisition:** Jens Bo Koudahl Conrad, Claus Lindbjerg Andersen.

**Investigation:** Jens Bo Koudahl Conrad, Tenna Vesterman Henriksen.

**Methodology:** Jens Bo Koudahl Conrad, Tenna Vesterman Henriksen, Jesper Berg Nors, Mai-Britt Worm Ørntoft, Mads Heilskov Rasmussen, Claus Lindbjerg Andersen.

**Resources:** Claus Lindbjerg Andersen.

**Supervision:** Tenna Vesterman Henriksen, Claus Lindbjerg Andersen.

**Validation:** Jens Bo Koudahl Conrad.

**Visualization:** Jens Bo Koudahl Conrad.

**Writing – original draft:** Jens Bo Koudahl Conrad, Tenna Vesterman Henriksen, Claus Lindbjerg Andersen.

**Writing – review & editing:** Tenna Vesterman Henriksen, Jesper Berg Nors, Mai-Britt Worm Ørntoft, Mads Heilskov Rasmussen, Nis Hallundbæk Schlesinger, Per Vadgaard Andersen, Kåre Andersson Gotschalck, Claus Lindbjerg Andersen.

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
