## [Decision Letter · Decision Letter 0]

17 Sep 2024

PONE-D-24-32619The role of renal and liver function in clinical ctDNA testingPLOS ONE

Dear Dr. Andersen,

Thank you for submitting your manuscript to PLOS ONE. After careful consideration, we feel that it has merit but does not fully meet PLOS ONE’s publication criteria as it currently stands. Therefore, we invite you to submit a revised version of the manuscript that addresses the points raised during the review process.

The authors need to improve and detail the methodology, including patients selection.

We look forward to receiving your revised manuscript.

Kind regards,

Elingarami Sauli, PhD

Academic Editor

PLOS ONE

3. In the online submission form you indicate that your data is not available for proprietary reasons and have provided a contact point for accessing this data. Please note that your current contact point is a co-author on this manuscript. According to our Data Policy, the contact point must not be an author on the manuscript and must be an institutional contact, ideally not an individual. Please revise your data statement to a non-author institutional point of contact, such as a data access or ethics committee, and send this to us via return email. Please also include contact information for the third party organization, and please include the full citation of where the data can be found.

Reviewers' comments:

Reviewer's Responses to Questions

**Comments to the Author**

1. Is the manuscript technically sound, and do the data support the conclusions?

Reviewer #1: No

Reviewer #2: Yes

2. Has the statistical analysis been performed appropriately and rigorously? 

Reviewer #1: I Don't Know

Reviewer #2: Yes

3. Have the authors made all data underlying the findings in their manuscript fully available?

Reviewer #1: Yes

Reviewer #2: Yes

4. Is the manuscript presented in an intelligible fashion and written in standard English?

Reviewer #1: Yes

Reviewer #2: Yes

5. Review Comments to the Author

Reviewer #1: The authors investigated whether cfDNA and ctDNA levels are influenced by impaired renal or liver function in patients undergoing curative-intent surgery for early-stage colorectal cancer. Their findings suggest that cfDNA and ctDNA levels remain stable across a broad spectrum of renal and liver function metrics.

However, several important limitations should be considered:

1. The study is focused on pre-operative cfDNA/ctDNA levels, but in clinical practice, post-operative ctDNA is more relevant. It cannot be assumed that post-operative cfDNA/ctDNA levels would remain unaffected by varying degrees of renal or liver function, particularly given the additional variable of DNA fragments released during surgery, which could influence ctDNA clearance in patients with impaired renal or liver function.

2. The method used to calculate eGFR is not clearly described, raising concerns about the accuracy and reliability of the renal function assessment.

3. The manuscript does not clarify whether patients had chronic kidney disease, defined as kidney damage or decreased kidney function persisting for three or more months. A single instance of slightly elevated creatinine does not accurately reflect true renal function.

4. Figure 1 indicates that only a small number of patients had creatinine levels above the normal reference range or an eGFR below the normal reference range. Drawing broad conclusions from such limited data may not be justified. The same concern applies to liver function tests, particularly serum bilirubin levels.

5. It would be beneficial to categorize patients with eGFR below 60 into subgroups based on the severity of renal impairment (e.g., eGFR 45-59, 30-44, etc.) to determine whether ctDNA levels are affected at specific eGFR threshold.

In summary, this manuscript does not offer any clinically valuable information.

Reviewer #2: The manuscript reports an interesting concept of impact of variation in renal and liver functions on the detection of ctDNA/cfDNA in patients with resectable colorectal cancer. The conclusion is that there is no such effect. I have the following comments.

1. The only major limitation in this study is that the majority of the participants have their renal/liver functions within reference range and hardly any participant has severe renal or liver dysfunction. This should be clearly highlighted as a limitation and reason for contradiction from the the pre-clinical and previously published reports.

2. It appears the bloods samples for ctDNA were taken both pre and post-surgery, but only pre-op data is presented. Did they compare post op ctDNA with post-op renal/liver functions?

6. PLOS authors have the option to publish the peer review history of their article (what does this mean? ). If published, this will include your full peer review and any attached files.

**Do you want your identity to be public for this peer review?** For information about this choice, including consent withdrawal, please see our Privacy Policy .

Reviewer #1: No

Reviewer #2: No

---

## [Author Response · Author response to Decision Letter 0]

9 Oct 2024

Reviewer #1: The authors investigated whether cfDNA and ctDNA levels are influenced by impaired renal or liver function in patients undergoing curative-intent surgery for early-stage colorectal cancer. Their findings suggest that cfDNA and ctDNA levels remain stable across a broad spectrum of renal and liver function metrics.

However, several important limitations should be considered:

1. The study is focused on pre-operative cfDNA/ctDNA levels, but in clinical practice, post-operative ctDNA is more relevant. It cannot be assumed that post-operative cfDNA/ctDNA levels would remain unaffected by varying degrees of renal or liver function, particularly given the additional variable of DNA fragments released during surgery, which could influence ctDNA clearance in patients with impaired renal or liver function.

Author Response: We appreciate this input from the reviewer. We agree that it is not possible to conclude that post-operative ctDNA levels are unaffected by the liver and renal function of the patients in this cohort. However, we believe that the complexity of the postoperative treatment regime makes it infeasible to draw conclusions about postoperative ctDNA levels with registry data. See our answer to comment number 2 from reviewer #2 for additional details.

2. The method used to calculate eGFR is not clearly described, raising concerns about the accuracy and reliability of the renal function assessment.

Author Response: We are thankful to the reviewer for this vigilant observation. The eGFR results were requisitioned from the regional databases of hospital laboratories using nationally standardized testing methodology. We have added a translated description of the eGFR assessment methodology and eGFR calculation formula in supporting information and revised the manuscript to refer to this [line 108. S1 File].

“In this study, we define a 30-day preoperative eGFR as the mean of all eGFR measurements conducted in the month leading up to the operation. The formulas used by the regional hospitals to calculate eGFR is available in S1 File.”

For ease of the reviewer we have also added them here:

Female

P-Creatinine ≤62 µmol/l:

eGFR = 144 x (P-Creatinine /(0.7 x 88.4))-0.329 x 0.993^age

P-Creatinine > 62 µmol/l:

eGFR = 144 x (P-Creatinine /(0.7 x 88.4))-1.209 x 0.993^age

Male

P-Creatinine ≤80 µmol/l:

eGFR = 141 x (P-Creatinine /(0.9 x 88.4))-0.411 x 0.993^age

P-Creatinine > 80 µmol/l:

eGFR = 141 x (P-Creatinine /(0.9 x 88.4))-1.209 x 0.993^age

3. The manuscript does not clarify whether patients had chronic kidney disease, defined as kidney damage or decreased kidney function persisting for three or more months. A single instance of slightly elevated creatinine does not accurately reflect true renal function.

Author Response: We thank the reviewer for bringing this lack of clarity to our attention. We have added a description of sample availability to the methods section [lines 106-112].

“In this study, we define a 30-day preoperative eGFR as the mean of all eGFR measurements conducted in the month leading up to the operation. The formulas used by the regional hospitals to calculate eGFR is available in S1 File. The median number of eGFR measurements per patient in the month up to operation was 2 (IQR 1-3). The 30-day preoperative eGFR was used to define whether a patient had low renal function or normal renal function. In accordance with Kidney Disease: Improving Global Outcomes (KDGIO) guidelines (https://kdigo.org) patients were categorized as having low renal function if their average eGFR was below 60 mL/min/1.73m2.

We acknowledge the reviewer’s concern in relation to assessing renal function based on a single eGRF measurement. The majority of patients had two or more eGFR measurements made in the 30-day timeframe before operation (n = 300, 62.4%). While we agree that the 30-day eGFR assessments based on a single measurement are less robust, we still believe that the affected patients contribute with important information. Furthermore, as the renal and fluid status measurements conducted in the run-up to operation were made at the treating clinician’s discretion, excluding patients with a single measurement would exacerbate the problem of confounding by indication. Due to these reasons, we abstain from excluding the patients.

In light of the points raised by the reviewer, we have added a section in the discussion to highlight this variance in sample numbers more clearly [line 223-229].

“Our study focused on laboratory test results in the short 30-day period before curative intended surgery as this period is the standard length of the Danish preoperative CRC cancer work-up program. However, this has the limitation of not meeting KDGIOs recommended 3-month sampling period for assessing long-term renal function. While most of our patients had multiple eGFR measurements in the 30-day period, a minor fraction had just a single eGFR measurement (n = 181 37.6%). These singular measurements provide important information, yet will not account for the patient’s biological variance in eGFR.”

4. Figure 1 indicates that only a small number of patients had creatinine levels above the normal reference range or an eGFR below the normal reference range. Drawing broad conclusions from such limited data may not be justified. The same concern applies to liver function tests, particularly serum bilirubin levels.

Author Response: We agree with the reviewer that our study offers limited information on patients with extremely low renal and liver function. However, it is highly relevant for the cfDNA/ctDNA community to get a better understanding of the relationship between the natural variation in renal function of CRC patients eligible for surgery and ctDNA test results. The cohort of this study is an unselected representative group of stage I-III CRC patients who are undergoing curative intended surgery. It would be possible to conduct a similar study on patients with end-stage renal and liver failure, but it would offer no actionable information for cancer patients undergoing curative intended treatments, as patients with extremely low renal and liver function are not candidates for surgery. We now address this important topic in the discussion section. [lines 202-204 and lines 236-237]

“Patients with much lower renal and liver function, than those included in this study, can be found in Denmark, but these would very rarely be candidates for curative intended surgery in the Danish healthcare system.”

“For now, it is encouraging that abnormal liver and renal function is of no great concern when conducting ctDNA testing, as the ctDNA clearance can be considered robust across a wide range of test results in CRC patients undergoing curative intended treatment.”

5. It would be beneficial to categorize patients with eGFR below 60 into subgroups based on the severity of renal impairment (e.g., eGFR 45-59, 30-44, etc.) to determine whether ctDNA levels are affected at specific eGFR threshold.

Author Response: We appreciate the reviewer’s warranted interest in this topic. We agree that a subgroup analysis of eGFR measurements is interesting. We originally chose not include these analyses in the manuscript, because there are very few patients in the severely impacted categories. However, based on the reviewer’s feedback we have now included the analyses in the supplementary materials (S1 and S2 Figures).

In summary, this manuscript does not offer any clinically valuable information.

Author Response: We respectfully disagree with this assertion. The field of ctDNA testing is currently undergoing rapid development. Uncovering possible sources off natural variation in ctDNA and cfDNA measurements remains an important concern for the field. Especially as attempts are made at potentiating the ctDNA levels in the bloodstream for sampling purposes.

Reviewer #2: The manuscript reports an interesting concept of impact of variation in renal and liver functions on the detection of ctDNA/cfDNA in patients with resectable colorectal cancer. The conclusion is that there is no such effect. I have the following comments.

1. The only major limitation in this study is that the majority of the participants have their renal/liver functions within reference range and hardly any participant has severe renal or liver dysfunction. This should be clearly highlighted as a limitation and reason for contradiction from the the pre-clinical and previously published reports.

Author response: We are thankful for the reviewer’s insightful and relevant suggestion. We do believe that our cohort is well suited to investigate whether ctDNA and cfDNA clearance inhibition occurs naturally in stage I-III CRC patients undergoing curative intended surgery. We agree wholeheartedly that it is important to communicate that our study does not provide much information on whether severe renal and liver dysfunction could inhibit cfDNA and ctDNA clearance. We have amended the discussion section to more clearly address the low prevalence of severe renal and liver function in our cohort. [lines 202-204]

“Patients with much lower renal and liver function, than those included in this study, can be found in Denmark, but these would very rarely be candidates for curative intended surgery in the Danish healthcare system.”

2. It appears the bloods samples for ctDNA were taken both pre and post-surgery, but only pre-op data is presented. Did they compare post op ctDNA with post-op renal/liver functions?

Author response: We appreciate the reviewer’s interest in preoperative ctDNA and postoperative renal/liver functions. Our study elects to focus on preoperative laboratory biomarkers and ctDNA tests. Preoperatively, patients have their liver and renal function systematically assessed as part of the clinical workup before surgery. In the postoperative situation, liver and renal function are much less systematic, meaning that indication, intervention, number of measurements and the timing vary considerably. A large fraction of these measurements was made after clinical intervention and outlier values were often quickly normalized. Accordingly, registry data of post-operative clinical renal and liver assessments are not an appropriate data source for making generalizable correlations between cfDNA/ctDNA and renal/liver function. In our view, elucidating the immediate post-operative ctDNA and cfDNA metabolism of cancer patients would require a clinical study specifically designed to prospectively and systematically gather the needed data (liver/renal/cfDNA/ctDNA measurements made at the same time-point), which is beyond the scope of this article.

The reviewer’s comment made us reflect on the methods section describing the blood samples available for ctDNA analysis. [Line 79 in original manuscript]

“As part of the study, patients had blood collected prior to and after surgery, with the aim to retrospectively measure cfDNA and ctDNA.”

As our study does not involve the postoperative samples, we have elected to remove the mentioning of these samples to avoid confusion. [Line 74]

Editorial office revision requests

and

Author response: We thank the editor for the resources provided and have used them to revise the manuscript to follow PLOS ONE’s style requirements: A list of changes is given below:

1. Edited headings to use sentence case, resized heads to 18pts for level 1 headings and 16pts for level 2 headings. Bolded all headings

2. Revised figure and table citations (example: figure 1 -> Fig 1. Supplementary table S1 -> S1 Table)

3. Revised filenames to follow guidelines

4. Unbolded figure captions except for figure title.

5. Changed title of Supplementary materials to “Supporting information” and updated contents to reflect manuscript edits.

6. Moved “supporting information section” to end of document

7. Removed the title of the “affiliations” section

8. Removed superfluous corresponding author information. Only email is listed now

9. Removed funding and support section

10. Removed declaration of competing interests section

Author response: We have elected to resubmit the manuscript in DOCX format. The original PDF submission was a mistake due to misunderstanding the author guidelines. We apologize for the mistake and are grateful for the opportunity to correct it.

3. In the online submission form you indicate that your data is not available for proprietary reasons and have provided a contact point for accessing this data. Please note that your current contact point is a co-author on this manuscript. According to our Data Policy, the contact point must not be an author on the manuscript and must be an institutional contact, ideally not an individual. Please revise your data statement to a non-author institutional point of contact, such as a data access or ethics committee, and send this to us via return email. Please also include contact information for the third party organization, and please include the full citation of where the data can be found.

Author response: We have edited the manuscript so that the contact point is The Department of Molecular Medicine at moma@rm.dk and included a description of how to use the mail to request data access.

“Because the data contains sensitive personal patient information, it cannot be made publicly available. Access to data requires that the data requestor (legal entity) enter into Collaboration and Data Processing Agreements, with the Central Denmark Region (the legal entity controlling and responsible for the data). Inquiries for access can be addressed to the Data Access Committee at Department of Molecular Medicine, Aarhus University Hospital (contact via moma@rm.dk).”

---

## [Decision Letter · Decision Letter 1]

3 Dec 2024

PONE-D-24-32619R1The role of renal and liver function in clinical ctDNA testingPLOS ONE

Dear Dr. Andersen,

Thank you for submitting your manuscript to PLOS ONE. After careful consideration, we feel that it has merit but does not fully meet PLOS ONE’s publication criteria as it currently stands. Therefore, we invite you to submit a revised version of the manuscript that addresses the points raised during the review process.

The authors should clearly compare post op ctDNA with post-op renal/liver functions, to show their difference with pre op data.

We look forward to receiving your revised manuscript.

Kind regards,

Elingarami Sauli, PhD

Academic Editor

PLOS ONE

Journal Requirements:

Reviewers' comments:

Reviewer's Responses to Questions

**Comments to the Author**

1. If the authors have adequately addressed your comments raised in a previous round of review and you feel that this manuscript is now acceptable for publication, you may indicate that here to bypass the “Comments to the Author” section, enter your conflict of interest statement in the “Confidential to Editor” section, and submit your "Accept" recommendation.

Reviewer #1: (No Response)

Reviewer #2: All comments have been addressed

2. Is the manuscript technically sound, and do the data support the conclusions?

Reviewer #1: No

Reviewer #2: Yes

3. Has the statistical analysis been performed appropriately and rigorously? 

Reviewer #1: I Don't Know

Reviewer #2: Yes

4. Have the authors made all data underlying the findings in their manuscript fully available?

Reviewer #1: Yes

Reviewer #2: No

5. Is the manuscript presented in an intelligible fashion and written in standard English?

Reviewer #1: Yes

Reviewer #2: Yes

6. Review Comments to the Author

Reviewer #1: This manuscript does not contribute any significant useful information to the literature as most patients had normal liver and renal function. Besides the timepoint of ctDNA/cfDNA measurement is clinically irrelevant.

Reviewer #2: The authors have addressed the concerns adequately. Though the impact of the findings are unlikely to be significant to current practice, the manuscript on its own is technically well written otherwise. I have no other issues with the manuscript.

7. PLOS authors have the option to publish the peer review history of their article (what does this mean? ). If published, this will include your full peer review and any attached files.

**Do you want your identity to be public for this peer review?** For information about this choice, including consent withdrawal, please see our Privacy Policy .

Reviewer #1: No

Reviewer #2: No

---

## [Author Response · Author response to Decision Letter 1]

17 Jan 2025

Editorial office revision requests

1. The authors should clearly compare post op ctDNA with post-op renal/liver functions, to show their difference with pre op data.

Author response: We concur that there is justified interest in exploring the correlation between renal/liver functions and cfDNA/ctDNA also in the postoperative setting. However, we believe there are serious limitations and challenges to using the data of this study for such a purpose.

Challenges in postOP analysis of the correlation between ctDNA and renal/liver functions:

As colorectal cancer (CRC) patients are often cured by surgery, most colorectal cancer patients have no ctDNA in the blood after surgery. Additionally, our cohort consists mainly of low-risk patients (pT1-3pN0, 63.6%). Of the 846 person cohort, only 62 (7.24%) have a positive ctDNA call in the postoperative sample. Furthermore, for the few patients who have residual disease after surgery, the initiation of adjuvant chemotherapy (ACT) impacts the residual tumor burden, leading to a lowering, and potentially elimination, of the ctDNA fraction in blood. Therefore, only blood samples collected prior to initiation of ACT are relevant for analysis of the correlation between ctDNA and renal/liver function.

Altogether, the number of patients for which we have suited samples with ctDNA detected before ACT initiation is very low (n = 59). This is too few for statistical analysis.

Challenges in postOP analysis of the correlation between cfDNA and renal/liver functions:

Postoperative cfDNA analyses are complicated by the temporary surgical trauma-induced surge in normal DNA circulating in the blood [1]. We have previously reported that the normal DNA level in the blood increases a median of 3.6-fold shortly after the surgical trauma. After this, the concentration slowly returns to normal level, which is typically reached around week four after surgery. Therefore, the correlation between cfDNA level and renal/liver function can only be studied in samples collected after week 4.

Also for cfDNA analysis, initiation of ACT is challenge. The ACT likely impacts the hematopoietic cellular turnover, and thereby the normal DNA level in blood. ACT is also likely to impact the renal/liver cfDNA clearance mechanisms. Therefore, it is futile to investigate the correlation between cfDNA and renal/liver functions after ACT initiation. In Denmark, the recommendation is to start ACT between week 4 and week 6, and the median ACT start date in our cohort is 31 days after surgery. Consequently, the window is very narrow for collection of samples that can be meaningfully used to assess the correlation between cfDNA and renal/liver function.

Challenges with PostOP analysis of laboratory biomarkers of renal/liver function:

There are also challenges related to the laboratory biomarkers in the postoperative setting. In the preoperative setting, patients have laboratory tests performed to map the baseline renal and liver function prior to deciding the treatment plan. The situation is very different in the immediate postoperative setting. There, renal/liver function is measured because the patient exhibits symptoms that compels a clinician investigate for treatment indication. Therefore, there is significant “confounding by indication” in the available data. Furthermore, as the laboratory test results are made by indication they are monitored closely and clinical intervention is immediately initiated for patients with deviant values. Effectively, the analysis by indication and the subsequent interventions make the laboratory test-results useless for analyses of any natural correlation between cfDNA and laboratory biomarkers of renal/liver function.

To illustrate the challenge, we have included a figure with serial potassium values from 6 patients measured over a period from operation to day 60. The oscillating potassium values are consistent with clinical intervention after deviant values were registered. Consequently, for a meaningful analysis of the correlation between cfDNA and renal/liver function, the cfDNA and renal/liver laboratory biomarker needs to be measured very closely in time.

Figure: line plots showing potassium levels for six patients in the cohort. Each line is a single patient’s potassium levels from day of surgery until 60 days after. The red dotted lines signify upper and lower reference range bounds.

We explored how many of our patients had blood samples available that fitted the above requirements (collected after week 4 and prior to initiation of ACT) and for which renal/liver function laboratory test results were available on same day or up to 5 days after. The upper time limit for cfDNA sample inclusion was 57 days after surgery, which is a 30 day time period from the end of week 4.

In total, we had 253 cfDNA samples four weeks after surgery and before initiation of ACT. Of these, 33 (13%) patients had laboratory markers taken in the valid timeframe, which is too few for sound statistical analysis. Properly addressing how postoperative renal and liver function impacts ctDNA and cfDNA metabolism will require a new prospective clinical trial, designed specifically for this purpose. Such a trial is beyond the scope of this article.

While we maintain that our dataset is suboptimal to address the question. We acknowledge that this question is of interest to the reviewer and the readership of PlosOne. Therefore, we have incorporated an analysis of postoperative cfDNA clearance in the supporting information section. This includes methodology, figures, full regression results and a discussion of the limitations and challenges involved in investigating the postoperative metabolism of cfDNA and ctDNA. We refer to S2 file in the revised manuscript [lines 195-197].

“In addition we conducted a supplementary investigation of cfDNA levels in the postoperative setting (S2 File) and found no significant association between postoperative cfDNA level and biomarkers of renal and liver function”

We hope that presenting this analysis will be of use to researchers wishing to investigate this subject in more detail in the future.

1. Henriksen TV, Reinert T, Christensen E, Sethi H, Birkenkamp-Demtröder K, Gögenur M, et al. The effect of surgical trauma on circulating free DNA levels in cancer patients—implications for studies of circulating tumor DNA. Molecular Oncology. 2020;14(8):1670-9.

---

## [Editor Report · Decision Letter 2]

29 Jan 2025

The role of renal and liver function in clinical ctDNA testing

PONE-D-24-32619R2

Dear Dr. Andersen,

We’re pleased to inform you that your manuscript has been judged scientifically suitable for publication and will be formally accepted for publication once it meets all outstanding technical requirements.

Kind regards,

Elingarami Sauli, PhD

Academic Editor

PLOS ONE
---

## [Editor Report · Acceptance letter]

PONE-D-24-32619R2

PLOS ONE

Dear Dr. Andersen,

I'm pleased to inform you that your manuscript has been deemed suitable for publication in PLOS ONE. Congratulations! Your manuscript is now being handed over to our production team.

Kind regards,

on behalf of

Dr. Elingarami Sauli

Academic Editor

PLOS ONE